# French General Practitioners’ Adaptations for Patients with Suspected COVID-19 in May 2020

**DOI:** 10.3390/ijerph20031896

**Published:** 2023-01-19

**Authors:** Aline Ramond-Roquin, Sylvain Gautier, Julien Le Breton, Yann Bourgueil, Tiphanie Bouchez

**Affiliations:** 1Département de Médecine Générale, University Angers, F-49000 Angers, France; 2Univ. Angers, Univ. Rennes, Inserm, EHESP, Irset (Institut de Recherche en Santé, Environnement et Travail)-UMR_S 1085, SFR ICAT, F-49000 Angers, France; 3Collège National des Généralistes Enseignants (CNGE), F-75000 Paris, France; 4Département de Médecine de Famille et de Médecine d’Urgence, Université de Sherbrooke, Sherbrooke, QC J1K 2R1, Canada; 5Inserm U1018, CESP, Primary Care and Prevention, University of Versailles Saint-Quentin, Paris-Saclay University, F-94807 Villejuif, France; 6Mission RESPIRE, EHESP-CNAMTS-IRDES–EA MOS 7348 EHESP, F-93210 Saint-Denis, France; 7Assistance Publique–Hôpitaux de Paris, Department of Epidemiology and Public Health, Hospital Raymond Poincaré, F-92380 Garches, France; 8Inserm, IMRB, CEpiA, University Paris-Est Créteil Val de Marne, F-94010 Créteil, France; 9Department of General Practice, Faculty of Health, University Paris-Est Créteil Val de Marne, F-94010 Créteil, France; 10French Society of General Medicine (SFMG), F-92130 Issy-les-Moulineaux, France; 11Institut Jean-François REY (IJFR), F-75010 Paris, France; 12Institute for Health Research and Documentation (IRDES), F-75010 Paris, France; 13Department of Education and Research in General Practice, RETINES, HEALTHY, Université Côte d’Azur, F-06000 Nice, France

**Keywords:** primary care, care organization innovation, COVID-19, interprofessional relations, pandemic, general practice, family practice

## Abstract

In France, towards the end of the first lockdown, COVID-19 management was largely transferred from hospitals to primary care. Primary care actors adapted their practices to ensure patients’ access to care, while limiting contamination. In this study, we aimed to identify patterns of adaptations implemented by French general practitioners (GPs) in May 2020 for outpatients with confirmed or suspected COVID-19, and factors associated with these adaptions. A French survey concerning care organization adaptations, and individual, organizational, and territorial characteristics, was sent to GPs. Data were analyzed by multiple correspondence analysis followed by agglomerative hierarchical clustering to identify GPs’ adaptation clusters. A multinomial logistic regression model estimated the associations between clusters and individual, organizational, and territorial factors. Finally, 3068 surveys were analyzed (5.8% of French GPs). Four GPs’ adaptation clusters were identified: autonomous medical reorganization (64.2% of responders), interprofessional reorganization (15.9%), use of hospital (5.1%), and collaboration with COVID-19 outpatient centers (14.8%). Age, practice type and size, and territorial features were significantly associated with adaptation clusters. Our results suggest that healthcare systems should consider organizational features of primary care to effectively deal with future challenges, including healthcare crises, such as the COVID-19 pandemic, but also those linked to epidemiologic and societal changes.

## 1. Introduction

In response to the severe acute respiratory syndrome-coronavirus 2 (SARS-CoV-2)-related pandemic, the French government implemented the first lockdown between the 17th of March 2020 and the 11th of May 2020 [1]. Leaving home was only allowed for a limited number of compelling reasons, leading to a major limitation of all activities in France. As in other countries worldwide, the aim of the lockdown was to contain the epidemic during this time when there was much uncertainty and a scarcity of critical material [2,3]. Initially during the lockdown, the management of COVID-19 (coronavirus disease 2019) was mainly supported by the hospital sector, but its limits were rapidly exceeded. In May 2020, the primary healthcare sector was solicited to provide local care for patients not requiring the technical platforms in hospitals. The primary healthcare professionals had to adapt to ensure patient care and to limit viral propagation. As seen in other countries [4,5,6], in France [7], teleconsultation became widespread, with increased payment options, and extensive communication concerning COVID. Moreover, in some regions in France [8], and in other countries [9], “COVID outpatient centers” were established: either by reorganizing existing healthcare structures or by creating new structures.

The French primary healthcare sector is mainly composed of private-sector healthcare professionals that are paid according to a fee-for-service model, financed by the national health insurance system. Since 2006, a gate-keeping system to regulate access to specialist care has been implemented in France [10]. During the same period, we also observed a trend towards group practices in primary care [11]. More recently, two main types of multidisciplinary primary care teams coordinated around a health project have emerged: independent multidisciplinary groups (1617 centers), composed of several independent healthcare professionals, and healthcare centers (428 centers), where healthcare professionals are employed. Currently, in France, about 15% of primary care professionals (depending on the profession) practice in these primary care teams. In 2016, a new territorial organization framework (“territorial and professional healthcare communities”) was introduced to encourage primary care actors to assume collective social responsibility in their region [12,13,14] This is quite similar to the primary care clusters in Wales [15]. Today, the French primary care sector is evolving towards more collective and integrative practices.

In France, the primary healthcare mobilization, during the COVID pandemic, was reported to be heterogeneous by some exploratory studies [16]. The French interdisciplinary primary care research network (“Assembler, Coordonner, COmprendre, Rechercher, Débattre en soin primaires [ACCORD]”) performed two national surveys to assess how primary care professionals, in particular general practitioners (GPs) adapted to the pandemic, and to identify the factors, particularly organizational factors, associated with these adaptions [17]. The first survey was performed at the start of the first lockdown, in March 2020, and the second at the end of the first lockdown, in May 2020. In the first survey, among other results, 70.9% of GPs that responded changed their practices in response to the pandemic: particularly those in multidisciplinary group practices [18]. This suggests that the GPs’ responses to the pandemic may depend on practice-related factors.

The objective of this study was to use data collected in the second survey, during May 2020, to identify adaptive patterns implemented by French GPs for managing outpatients with confirmed or suspected COVID-19 infections. The study also aimed to identify individual, organizational, and territorial factors associated with these adaptions.

## 2. Materials and Methods

### 2.1. Study Design

The study used data collected during the second survey performed by the ACCORD network between the 7th and 20th of May 2020, at the end of the first lockdown. The survey was designed by an interdisciplinary research group (that included experts from general practice, public health, health services research, advanced nursing, midwifery, sociology, and representatives of multidisciplinary practices). The survey was based on clinical practice, data from the literature, and the results of the first survey performed at the beginning of the first lockdown [18]. The survey was reviewed by seven primary healthcare experts from the research group and tested in a pilot study performed by members of the survey team. This pilot study evaluated the understanding and readability of the questions and of the answers’ modalities. The research group reviewed the questionnaire according to the pilot study’s feedback until consensus was reached. The final questionnaire was distributed, by email (using the Limesurvey tool), to a list of approximately 25,000 GPs and a further 4436 GPs that completed the first survey and who agreed to complete the second. The list of GPs was obtained through partnerships with several general practice organizations, listed in the Appendix A. Moreover, the survey was advertised on social media by GPs. GPs were informed of the study objectives, design, methods, funding sources, and the potential benefits. GPs that agreed to participate provided informed consent before completing the survey.

### 2.2. Data Collection

The data were collected using the 63-item survey. The survey was divided into two parts. The first part explored seven adaption domains: activity, tests and prescriptions, occupational health, patients with suspected or confirmed COVID-19, nursing home residents, vulnerable patients, and territorial partnerships. The second part collected data about the participating GPs and the organizational and territorial characteristics of their practices. The individual data collected about the GPs included their sex, age, teaching activities, the numbers of patients that they usually and currently consult, the numbers of their patients hospitalized and those who died due to COVID-19, their perspective on being at risk of severe COVID-19, and their fear of the emerging severe acute respiratory syndrome coronavirus 2 (SARS-CoV-2). The organizational characteristics data concerned the type of practice (either works alone, in a monodisciplinary group practice, in an independent multidisciplinary group, or in a healthcare center), and the size of the practice. While the territorial characteristics data comprised the practice locations, the relations with the hospitals, and local partnerships.

In this study, we focused on assessing the GPs responses to the patients with suspected or confirmed COVID-19 adaption domain of the survey. The 13 items explored within this domain are listed in Table 1.

### 2.3. Eligibility Criteria

The data from GPs were included in the study if the GP completed the first part of the survey and the main variables of the second part: their sex and age, as well as the organizational and territorial characteristics of their practice. Duplicates were identified, and excluded, using the access numbers to the survey for responders to the previous survey, the GP’s registration numbers in the national physicians’ directory, or their email addresses (duplicates appeared due to the same participant completing the questionnaire more than once). Only data from GPs working in metropolitan France were included in this study due to significant differences between metropolitan France and overseas territories in terms of the health situation and the health care system organization.

### 2.4. Data Analyses

To analyze the data, we initially performed a univariate analysis. Then, to reduce the dimensionality and to eliminate redundancy across the 13 categorical elements, we performed a multiple correspondence analysis. The multiple correspondence analysis, using scree plots and Kaiser criterion, was performed to determine the smallest number of dimensions required to explain the largest portion of variance [19]. The multiple correspondence analysis coordinates were then combined by agglomerative hierarchical clustering, using the Ward minimum-variance linkage method [20]. Clusters were then selected to maximize inter-cluster inertia and with parsimony to generate a typology of GPs’ adaption patterns.

Finally, a multinomial logistic regression model was used to estimate associations between the dependent variable, the newly created typology (an unordered multicategory outcome), and the independent variables: the individual, organizational, and territorial factors. For organizational factors, the independent multidisciplinary groups and healthcare centers were merged into one modality: “multidisciplinary practice”. Moreover, since practice type and size were expected to be collinear and considered to be very relevant for the study, a composite variable was created: “practice type and size”. This composite variable had five modalities: (i) alone, (ii) monodisciplinary group practice (2–5 professionals), (iii) monodisciplinary group practice (≥6 professionals), (iv) multidisciplinary practice (2–19 professionals), and (v) multidisciplinary practice (≥20 professionals). Variables were not selected during the statistical modelling since all variables introduced were of interest. The likelihood-ratio chi-square, score, and Wald tests were used to assess the fit of the multivariate model [21]. The SAS software (version 9.4) was used for all analyses and significance was set at 5%.

## 3. Results

### 3.1. Sample Characteristics

Finally, the analysis included 3068 surveys. The study flow chart is shown in Figure 1. The responding GPs had a mean age of 46.6 years and 55.2% were women (Table 2). Moreover, 1330 GPs (43.4%) worked in monodisciplinary group practices. The study sample represented 5.8% (3068 of 53,339) of GPs practicing in France [22]. Compared to all French GPs, the responders were younger with 37.7% younger than 40 years compared to 17.0% younger than 40 years in all French GPs. The responders were also more frequently women (55.2% of the survey responders versus 44.2% of all French GPs) [22] and worked less frequently alone (15.5% of the survey responders vs. 39.0% of all French GPs) [9] (Table 2). Teaching GPs were overrepresented (68.3 vs. 20% of all French GPs). All metropolitan French regions were represented in the study (Appendix A). Concerning COVID, 2515 GPs (82.0%) had seen at least one patient for COVID-19-related reasons in the 7 days before completing the survey. COVID-19-related activity represented >10% of the clinical activity of 602 GPs (19.6%). Regarding the number of each GP’s regular patients who died due to COVID-19, the very asymmetric distribution of the data led to the recoding of the data into 3 modalities: 0 patient (71.6% of the sample); 1 or 2 (21.3%); 3 or more patients (7.1%).

### 3.2. Typology of General Practioners’ Adaptions for Patients with Suspected or Confirmed COVID-19

The multiple correspondence analysis and hierarchical clustering identified a 4-cluster typology that was clinically meaningful (Table 3). The autonomous medical reorganization cluster (Cluster 1) was the largest, comprising 1970 GPs (64.2%). In this cluster, GPs adapted their offices to safely welcome all patients, and continued to follow infected patients. In the interprofessional reorganization cluster (Cluster 2), with 488 GPs (15.9%), GPs dedicated specific time slots and spaces for patients with suspected COVID-19. Moreover, they collaborated with other healthcare professionals in their practices to follow patients with COVID-19. The use of hospital cluster (Cluster 3) comprised 156 GPs (5.1%). These GPs sent their patients with suspected COVID-19 to the hospital and on occasion delegated follow-up to the hospital sector. Finally, the COVID-19 outpatient centre cluster (Cluster 4) comprised 454 GPs (14.8%). These GPs often collaborated with centers dedicated to the management of COVID-19 patients.

### 3.3. Multivariate Modelling of General Practitioners’ Adaptations for the Management of Patients with Suspected or Confirmed COVID-19

The autonomous medical reorganization cluster (Cluster 1) was chosen as the reference modality for the dependent variable. The regression analysis was performed using 3020 surveys, 48 surveys were excluded due to missing data. The objective of the analysis was to predict the probability of a GP belonging to one of the three minority clusters (Clusters 2–4) relative to the reference cluster (Cluster 1). According to the likelihood-ratio chi-square, score, and Wald tests the multinomial logistic regression model was well fitted (Table 4).

Age was found to be a significant factor. Indeed, GPs younger than 40 years of age were less likely to use the hospital (adjusted odds ratio ([aOR] = 0.43 [0.25–0.74]) or to collaborate with COVID-19 outpatient centers (aOR = 0.60 [0.46–0.79]), compared to GPs aged 40 to 54 years. None of the other individual factors were significantly associated. Compared with GPs working in monodisciplinary group practices (2–5 professionals), GPs working alone were more frequently in the use of hospital cluster (Cluster 3; aOR = 1.98 [1.27–3.11]). GPs working in monodisciplinary group practices (≥6 professionals), multidisciplinary practices, and those members of a territorial and professional healthcare community were more frequently in the interprofessional reorganization (Cluster 2) or COVID-19 outpatient centre cluster (Cluster 4; aOR >1 for all). Lastly, GPs who considered hospitals as supportive for care organization were more frequently in the use of hospital cluster (Cluster 3; aOR = 1.88 [1.32–2.68]) and COVID-19 outpatient center cluster (Cluster 4; aOR = 1.84 [1.46–2.30]).

## 4. Discussion

Our results suggest that French GPs largely adapted their practices and organizations to the first COVID-19 lockdown. We identified four categories of GPs based on how they organized their work to manage patients with suspected/confirmed COVID-19. Organization features, among the factors assessed, were more frequently significantly associated with the GPs’ adaptation patterns.

This study was conducted using an interdisciplinary and participative approach, offering a rare insight into GPs’ adaptations at the beginning of the pandemic in France. More than 3000 GPs, throughout metropolitan France, representing more than 5% of all French GPs participated in the study. The participants were younger, more frequently women, and worked less frequently alone compared with all French GPs. Teaching GPs were also overrepresented. These characteristics correspond to a sub-population of French GPs particularly involved in innovative practices and cooperative working practices. Moreover, since GPs that teach influence the practices of future GPs, our results may offer a vision of the future primary care workforce in France [23]. Our study may have overestimated the percentage of GPs who adapted their work organization due to social desirability bias. Beyond the crude estimations, we identified important associations between practice-related factors and GPs’ adaptations during the COVID-19 pandemic: an area of research that has not yet been extensively studied.

In our survey, most GPs reorganized their work to ensure care and follow-up for patients with suspected COVID-19. They introduced dedicated time slots and physical spaces, and often performed teleconsultations. Teleconsultations were not often used prior to the pandemic but became widespread during the pandemic [24,25,26,27]. GPs may consider teleconsultations as an alternative to classical consultations to maintain access to healthcare while limiting the risk of contamination [28,29,30]. However, teleconsultations can be limited by the quality of the consultation, particularly for vulnerable people [31,32]. Moreover, teleconsultations can be difficult to perform for healthcare professionals [33] and are not always considered acceptable by patients [34]. It is noteworthy that the use of teleconsultations decreased after the first wave of the pandemic [35,36,37].

A significant proportion of GPs relied on COVID-19 outpatient centers. These centers were established to ensure access to healthcare while implementing protective measures for patients and healthcare professionals [8,38]. These centers were implemented using various modalities in different countries [39,40,41]. In France, some centers were closed after only a few months, whereas others evolved to offer diagnostic tests and vaccinations. Overall, centers organized within pre-existing primary care structures seemed to be more efficient in adapting their resources to the pandemic and healthcare needs (e.g., pandemic waves and vaccination strategies) [16]. They also tended to exist for longer.

Younger GPs used hospitals and COVID-19 outpatient centers less frequently. This suggests either a lower capacity to collaborate with territorial partners, or an increased capacity to use internal resources. This latter is likely considering that significantly more GPs younger than 40 years reported internal reorganization, compared to older GPs in our study. Moreover, younger French GPs are reported to be extensively trained for interprofessional collaboration and favor new practice modes within interprofessional teams [42,43]. In contrast, older GPs more often used hospitals resources. When adjusting for other variables, this association remained just borderline significant, suggesting some confusion bias. Specifically, this observation may be (at least partly) explained by the fact that older GPs more often work alone.

Our study suggests that the organization features (practice type and size) were the factors more significantly associated with the GPs’ capacity for adaptation. GPs working alone relied more on hospitals, suggesting that they had fewer resources to reorganize their work within the primary care framework. The lower capacity of GPs working in small practices to adapt was also identified in the United States [44] while task changes were larger in primary care practices employing a wider range of professionals, based on a large survey undertaken in 2020–2021 in 38 countries [45]. In France, a poor territorial primary care structure and the proactive positioning of some hospitals, during lockdown, may have promoted the use of hospital resources for patients without severe symptoms. A hospital-based strategy was favored in some countries with a high epidemic burden, e.g., South Africa and Egypt where the integration between primary care and public health was insufficient [5].

In contrast, GPs working in large practices (≥6 physicians) and within interprofessional teams collaborated more frequently with COVID-19 outpatient centers and with other professionals within coordinated teams. These findings suggest that larger practices are more conducive to collaborative work: geographical, organizational, and social proximity with other healthcare professionals favors collaboration [46]. This increased collaboration may also have been facilitated by sharing of resources (financial means, protocols, workforce), prior to the pandemic, that could be rapidly mobilized during the pandemic. The role of interprofessional practice, in France, during the pandemic has been documented in some qualitative studies [16,47]. Moreover, we have reported that interdisciplinary practice played an essential role for assuring care continuity for frail patients during the pandemic [48]. However, data on interprofessional practice during the COVID-19 pandemic are scarce, as are guidelines to indicate the role and recommended reorganization of primary care during a pandemic [6]. Interprofessional work and collaboration between primary care and public health professional was highlighted during previous epidemics [49,50] and may be critical for future healthcare crises [51].

Our findings suggest that the adaptability of primary care professionals is mostly influenced by practice organization and professional habits that existed before the pandemic. Consequently, the intervening COVID-19 pandemic did not result in organizational rupture but in a strengthening of existing structures, partnerships, and dynamics. These hypotheses need to be thoroughly investigated to better understand how primary care actors can adapt to crises or cope with other challenges. A more integrated primary care offer, adapted to the population in a specific territory, may be more resilient in crises. To assess this, the efficiency of different primary care organization models, their acceptability to the target populations, and their sustainability (considering the territorial context), needs to be evaluated. This would inform policymakers and stakeholders involved in planning primary care workforce development. Such a research program would require an ecosystem that continuously supports primary care research with a dedicated information system.

## 5. Conclusions

In conclusion, our study provides new insights into the mobilization of French GPs at the start of the COVID-19 pandemic, during the first lockdown in France. We identified four GP clusters according to how they organized their work to manage patients with suspected/confirmed COVID-19, by relying or not, on various partners. Moreover, we highlight organizational factors that impacted the capacity of GPs to cope with the health crisis. Our results suggest that healthcare systems should rely more on primary care actors, taking into consideration the importance of organizational features to be more effective during future challenges, including those emerging during healthcare crises, such as the COVID-19 pandemic, but also those linked to demographic, epidemiologic, and societal changes.

## Figures and Tables

**Figure 1 ijerph-20-01896-f001:**
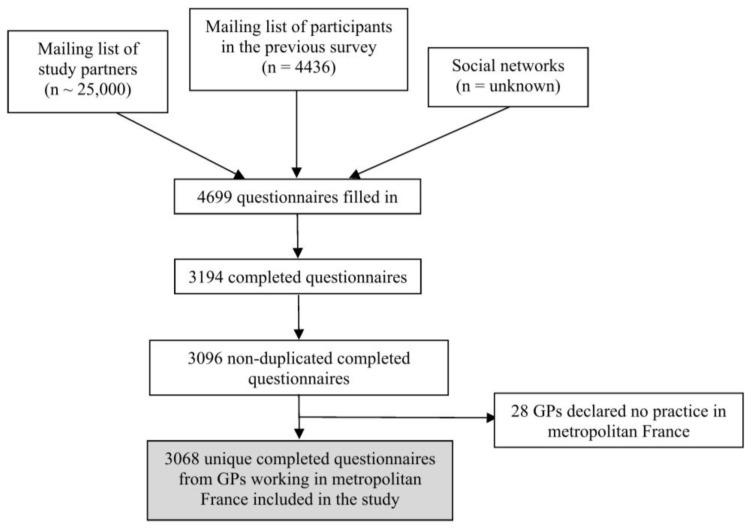
The study flow chart.

**Table 1 ijerph-20-01896-t001:** A list of the 4 sub-domains and 13 items to describe the adaption of general practitioners (GPs) for patients with suspected or confirmed COVID-19.

The 4 Sub-Domains	The 13 Items in the Patients with Suspected or Confirmed COVID-19 Domain of the Survey
Intra-practice adaptations for patients with suspected COVID-19	Dedicated time slots
	Dedicated spaces
	Home visits
	Teleconsultation
Outside-practice referrals for patients with suspected COVID-19	Systematic referral to a COVID-19 outpatient centre (created or reorganized)
	Systematic referral to the hospital
Territorial cooperation	Participation in the activity of a COVID-19 outpatient centre (created or reorganized)
	Seeing patients living in the practice catchment area but not registered with the responding GP
	GPs from the same catchment area welcomed in the practice for consultations
Follow up of patients with confirmed COVID-19 without severity criteria	Follow up by a GP within the practice
	Follow up by another healthcare provider within the practice
	Follow up by an administrative staff member of the practice
	Follow up by the hospital

**Table 2 ijerph-20-01896-t002:** The main characteristics of responding general practitioners (n = 3068) compared with all general practitioners practicing in metropolitan France.

Characteristics	Responding GPs, n = 3068	All GPs in Metropolitan France ^1^, n = 53,339
Sex, n (%)		
Women	1694 (55.2)	23,576 (44.2)
Men	1374 (44.8)	29,763 (55.8)
Age group (in years), n (%)		
<40	1155 (37.7)	9068 (17.0)
(40–54)	879 (28.7)	15,255 (28.6)
≥55	1023 (33.3)	29,016 (54.4)
*Missing data*	*11*	
Type of practice, n (%)		
Alone	474 (15.5)	20,802 (39.0) ^2^
Monodisciplinary group practice	1330 (43.4)	
Multidisciplinary practice	1264 (41.2)	
incl. Independent multidisciplinary group	1113	
incl. Care centers	135	
*incl. Missing data*	*16*	

GPs, general practitioners. ^1^ Data from the French health insurance system (CNAMTS) (2019) [22]. ^2^ Data from the French direction of research, studies, evaluation, and statistics (DREES) (2019)–No data was available for the other types of practices [11].

**Table 3 ijerph-20-01896-t003:** Organization adaptations by general practitioners (GPs) for patients with suspected or confirmed COVID-19 at the end of the first lockdown: typology in four clusters.

Variables	All, n = 3068	Cluster 1, n = 1970 (64.2%)	Cluster 2, n = 488 (15.9%)	Cluster 3, n = 156 (5.1%)	Cluster 4, n = 454 (14.8%)
**Active variables, n (%)**					
Intra-practice adaptations for patients with suspected COVID-19					
Dedicated time slots	1952 (63.6)	1347 (68.4)	359 (73.6)	86 (55.0)	160 (35.2)
Dedicated spaces	1540 (50.2)	956 (48.5)	335 (68.6)	81 (52.0)	168 (37.0)
Home visits	368 (12.0)	232 (11.8)	65 (13.3)	34 (21.8)	37 (8.1)
Teleconsultation	2430 (79.2)	1599 (81.2)	388 (79.5)	117 (75.0)	326 (71.8)
Outside-practice referrals for patients with suspected COVID-19					
Systematic referral to a COVID-19 outpatient centre (created or reorganized)^1^	394 (12.8)	21 (1.1)	25 (5.1)	26 (16.7)	322 (70.9)
Systematic referral to hospital ^1^	105 (3.4)	0	0	103 (66.0)	2 (0.4)
Territorial cooperation					
Participation in the activity of a COVID-19 outpatient centre (created or reorganized)	541 (17.6)	140 (7.1)	25 (5.1)	19 (12.2)	357 (78.6)
Seeing patients living in the practice catchment area but not registered with the responding GP	2304 (75.1)	1420 (72.1)	393 (80.5)	113 (72.4)	378 (83.3)
GPs from the same catchment area welcomed in the practice for consultations	156 (5.1)	2 (0.1)	17 (3.5)	7 (4.5)	130 (28.6)
Follow-up of patients with confirmed COVID-19 without severity criteria					
Follow-up by a GP within the practice	2921 (95.2)	1970 (100.0)	370 (75.8)	142 (91.0)	439 (96.7)
Follow-up by another healthcare provider within the practice	470 (15.3)	1 (0.1)	404 (82.8)	14 (9.0)	51 (11.2)
Follow-up by an administrative staff member of the practice	38 (1.2)	0	32 (6.6)	0	6 (1.3)
Follow-up by the hospital	65 (2.1)	0	0	63 (40.4)	2 (0.4)
**Attributed variables, n (%)**					
Individual factors					
Age					
<40 years	1155 (37.8)	794 (40.4)	187 (38.4)	24 (15.6)	150 (33.1)
(40–54)	879 (28.8)	523 (26.6)	154 (31.6)	41 (26.6)	161 (35.5)
≥55 years	1023 (33.5)	646 (32.9)	146 (30.0)	89 (57.8)	142 (31.3)
Male sex	1374 (44.8)	855 (43.4)	242 (49.6)	91 (58.3)	186 (41.0)
Teaching activities ^2^	2090 (68.1)	1286 (65.3)	372 (76.2)	112 (71.8)	320 (70.5)
Usual daily clinical activity (before the COVID-19 pandemic)					
<20 patients/day	404 (13.2)	252 (12.9)	73 (15.0)	23 (15.0)	56 (12.4)
20 to 29 patients/day	2341 (76.6)	1507 (76.8)	366 (75.0)	113 (73.9)	355 (78.4)
≥30 patients/day	310 (10.1)	202 (10.3)	49 (10.0)	17 (11.1)	42 (9.3)
Being at risk of severe COVID-19 ^3^	444 (14.5)	277 (14.1)	79 (16.2)	33 (21.2)	55 -12.1)
Fear of SARS-CoV-2 ^4^	412 (13.4)	262 (13.3)	57 (11.7)	27 (17.3)	66 (14.5)
At least one patient seen for a COVID-19-related reason (in the last 7 days)	2515 (82.0)	1634 (82.9)	375 (76.8)	129 (82.7)	377 (83.0)
>10% of activity linked to COVID-19 (in the last 7 days) ^5^	602 (19.6)	392 (19.9)	87 (17.8)	24 (15.4)	99 (21.8)
Number of patients who died of COVID-19 among regular patients					
none	2182 (71.6)	1388 (70.9)	339 (69.8)	112 (73.2)	343 (76.1)
1–2 patients	648 (21.3)	425 (21.7)	110 (22.6)	30 (19.6)	83 (18.4)
>2 patients	219 (7.2)	146 (7.5)	37 (7.6)	11 (7.2)	25 (5.5)
Practice-related factors, n (%)					
Type of practice					
Alone	474 (15.5)	331 (16.9)	33 (6.8)	48 (31.2)	62 (13.7)
Monodisciplinary group practice	1331 (43.6)	957 (48.7)	119 (24.6)	58 (37.7)	197 (43.6)
Multidisciplinary practice: independent multidisciplinary group	1113 (36.5)	599 (30.5)	302 (62.5)	44 (28.6)	168 (37.2)
Multidisciplinary practice: healthcare centre	135 (4.4)	77 (3.9)	29 (6.0)	4 (2.6)	25 (5.5)
Size of practice					
Alone	474 (15.5)	331 (16.9)	33 (6.8)	48 (31.2)	62 (13.7)
2–5 professionals	1283 (41.8)	944 (47.9)	113 (23.2)	51 (32.7)	175 (38.5)
6–19 professionals	1104 (36.0)	605 (30.7)	282 (57.8)	51 (32.7)	166 (36.6)
≥20 professionals	206 (6.7)	89 (4.5)	60 (12.3)	6 (3.8)	51 (11.2)
Territorial factors, n (%)					
Member of a territorial health organization ^6^	923 (30.1)	527 (26.8)	196 (40.1)	31 (19.9)	169 (37.2)
Hospital considered as support for care organization	797 (26.0)	462 (23.5)	122 (25.0)	56 (35.9)	157 (34.6)

^1^ “Often” or “Always” (clustered as “Yes”) versus “Never” or “Rarely” (clustered as “No”). ^2^ University internship supervisors. ^3^ Self-assessments of the risk of severe COVID-19 (“Are you at risk of severe COVID-19?” yes versus no). ^4^ “Are you afraid of being infected?” Answers “Not at all” or “Slightly” (clustered as “No”) versus “Afraid” or “Very afraid” (clustered as “Yes”). ^5^ Estimation of the percentage of patients seen for a COVID-19-linked reason (e.g., suspected, follow-up, contact case) relative to the whole activity in the last 7 days. ^6^ Participation in a territorial and professional healthcare community.

**Table 4 ijerph-20-01896-t004:** General practitioners’ adaptations for patients with suspected or confirmed COVID-19 at the end of the first lockdown: results of the multinomial logistic regression model.

Variables	Cluster 2, n = 488 (15.9%) Interprofessional Reorganization Cluster		Cluster 3, n = 156 (5.1%) Use of Hospital Cluster		Cluster 4, n = 454 (14.8%) COVID-19 Outpatient Center Cluster	
	aOR (95% CI)	*p* -Value	aOR (95% CI)	*p*-Value	aOR (95% CI)	*p*-Value
**Individual factors**						
Age						
<40 years	0.81 (0.62–1.06)	NS	0.43 (0.25–0.74)	≤0.01	0.60 (0.46–0.79)	≤0.01
(40–54)	Ref.		Ref.		Ref.	
≥55 years	0.81 (0.61–1.07)	NS	1.50 (0.98–2.28)	≤0.05	0.80 (0.61–1.05)	NS
Male sex	1.22 (0.98–1.52)	NS	1.27 (0.89–1.83)	NS	0.85 (0.68–1.07)	NS
Teaching activities ^1^	1.32 (1.02–1.70)	≤0.05	1.08 (0.72–1.62)	NS	1.05 (0.82–1.34)	NS
Usual daily clinical activity (before the COVID-19 pandemic)						
Teaching activities ^1^						
<20 patients/day	1.42 (1.05–1.92)	≤0.05	0.98 (0.59–1.62)	NS	0.95 (0.68–1.31)	NS
20 to 29 patients/day	Ref.		Ref.		Ref.	
≥30 patients/day	0.93 (0.65–1.33)	NS	0.80 (0.46–1.38)	NS	0.88 (0.61–1.26)	NS
Being at risk of severe COVID ^2^	1.24 (0.91–1.67)	NS	0.96 (0.61–1.50	NS	0.83 (0.60–1.15)	NS
Fear of SARS-CoV-2 virus ^3^	0.92 (0.66–1.27)	NS	1.30 (0.83–2.04)	NS	1.27 (0.94–1.72)	NS
Number of patients who died of COVID-19 among regular patients						
None	Ref.		Ref.		Ref.	
1–2 patients	1.04 (0.80–1.36)	NS	1.10 (0.70–1.72)	NS	0.98 (0.75–1.27)	NS
>2 patients	1.05 (0.81–1.36)	NS	1.41 (0.93–2.13)	NS	0.92 (0.71–1.19)	NS
**Practice-related factors**						
Type and size of practice						
Alone	0.90 (0.59–1.38)	NS	1.98 (1.27–3.11)	≤0.01	0.99 (0.71–1.38)	NS
Monodisciplinary group practice with 2–5 professionals	Ref.		Ref.		Ref.	
Monodisciplinary group practice with ≥6 professionals	2.11 (1.31–3.41)	≤0.01	1.84 (0.94–3.63)	NS	1.69 (1.11–2.58)	≤0.05
Multidisciplinary practice with 2–19 professionals	3.97 (3.05–5.18)	≤0.01	1.43 (0.92–2.23)	NS	1.25 (0.96–1.61)	NS
Multidisciplinary practice with ≥20 professionals	5.50 (3.65–8.27)	≤0.01	1.03 (0.36–2.97)	NS	2.82 (1.87–4.25)	≤0.01
**Territorial factors**						
Member of a territorial and healthcare organization ^4^	1.36 (1.09–1.70)	≤0.01	0.70 (0.46–1.07)	NS	1.57 (1.25–1.96)	≤0.01
Hospital considered as support for care organization	1.16 (0.91–1.49)	NS	1.88 (1.32–2.68)	≤0.01	1.84 (1.46–2.30)	≤0.01

aOR, adjusted odds ratio; CI, confidence interval; NS, not significant; SARS-CoV-2, severe acute respiratory syndrome coronavirus 2. ^1^ University internship supervisors. ^2^ Self-assessments of the risk of severe COVID-19 (“Are you at risk of severe COVID-19?” yes versus no). ^3^ “Are you afraid of being infected?” Answers “Not at all” or “slightly” (clustered as “No”) versus “Afraid” or “Very afraid” (clustered as “Yes”). ^4^ Participation in a territorial and professional healthcare community.

## Data Availability

The data underlying this article will be shared on reasonable request to the corresponding author.

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
