# Peer review of "French General Practitioners’ Adaptations for Patients with Suspected COVID-19 in May 2020"

_ijerph, 2023, doi:10.3390/ijerph20031896_

Round 1

Reviewer 1 Report

This is a clear account of a well-constructed survey of metropolitan French GPs during the first lockdown of COVID-19 between 17 March 2020 and 11 May 2020 to determine the result of the lockdown on their practice and on the ways in which they attended to patients.

There are, however, important details that have been left out, including some missing references, and a few small errors to correct. These details, references and the errors are noted in the line by line suggested edits to follow. 

Once these corrections are made, this study represents one that could be easily replicated because of the care taken in devising and implementing the study as well as conducting the analysis.

Line by line suggested edits:

32 Change “~5%” to “5.8%”. The rest of the percentages quoted are exact. This one should be too.

44 Although, at this time, the world understands what a COVID lockdown represents, for future generations, please add information indicating what is COVID and why there would be a lockdown. Please reference this as well.

89 Please indicate the disciplines represented in this interdisciplinary research group.

91-92 How were these seven primary healthcare experts chosen and contacted. What was their reason for participating in the review?

92 Please provide some details about the pilot study.

101 How was the survey constructed? What previous surveys influenced the design of this survey?

107 Change “patients that usually” to “patients that they usually”.

110 When additional detail is added about COVID in line 44, be sure to include that COVID-19 is SARS-CoV-2.

120  Delete “GP, general practitioner.” This information is unnecessary.

124 Please provide information regarding why there might be duplicates.

127-128 Please state why only data from GPs working in metropolitan France were included.

133 Please provide a recent reference to scree plots and Kaiser criterion. 

134 Change “explains” to “explain”.

136 Please provide a recent reference to the Ward minimum-variance linkage method.

150 Please provide a recent refence for the Wald tests.

161 To correspond with the data in Table 2 (and to add up to 100%), please change “44.2%” to “44.8%”. Regarding working less frequently alone, looking at Table 2, 15.5% represents those GP who worked alone, not those who didn’t work alone. As well, the 15.5% versus 39% is between those who answered the survey and all GPs in France rather than those who didn’t work alone and those who did. If the authors want to make this comparison with all French GPs, rather than in relation to whether those surveyed in the study work alone, then they must state this is what they are comparing. If the point they want to make is that most metropolitan French GPs who don’t work alone were those that they surveyed they must add 43.4% to 41.2% to indicate that 84.6% did not work alone.

163 If all French regions were represented how does this relate to the point made in lines 127-128 that only those GPs working in metropolitan France were included?

Table 3

Change “[40-55[“ to “[40-55]”

Change “≥55” to “>55”.

It is interesting that the largest percentage of GPs >55 were the ones who used the hospital cluster (Cluster 3). Is it a greater benefit to be able to use the hospital cluster and is this why they may have selected this type of practice? Please comment on this in the manuscript.

Why do the authors make a point of differentiating COVID-19 from SARS-CoV-2 in their list of variables?

Was “>2” the only option on the survey for the number of patients who died if the answer was > 2? Is it possible that some GPs had a significantly more deaths than two? If so, how was this captured in the survey? If it was not, why was this not considered important? Please explain in the manuscript.

Again, it is noted that those GPs in Cluster 3 had the highest percentage related to those who practiced alone in comparison with the other clusters. Is this because they were generally older than the GPs in the other clusters and older GPs were more likely to work alone?

191 Delete “GPs, general practitioners; SARS-COV-2, severe acute respiratory syndrome coronavirus 2.” This information is unnecessary.

192 Given that there was a range that GPs could select from, what does the percentage indicated represent? Is it GPs who selected anything other than “Never”? Please explain in the manuscript.

195 Given that there was a range that GPs could select from, what does the percentage indicated represent? Is it the GPs who selected anything other than “Not at all”? Please explain in the manuscript.

213 Also note that these GPs were more frequently >55.

Table 4

Change “[40-55[“ to “[40-55]”

Change “≥55” to “>55”.

227 Change “. Level” to “Level”. Given that there was a range that GPs could select from, what does the percentage indicated represent? Is it the GPs who selected anything other than “Not at all”? Please explain in the manuscript.

266-272 The authors have commented on younger GPs but have not made a general comment on those mid career or those who are older GPs. Mention should be made of results associated with them as well. 

275-276 It may also suggest that they needed few resources in working alone because they had more hospital privileges than other GPs—there was no need for duplication. Please comment on this in the manuscript.

Supplementary material Table S2

Rather than putting “GPs, general practitioners” at the bottom of the table, in the title, change “general practitioners’” to “general practitioners’ (GPs)”.

References

Generally, remember that MDPI journals require short forms for journal names rather than the full name of the journal. Some of the references need to have the journal title reduced to its short form.

9. This is an incomplete reference. What type of document is this?

17. This is an incomplete reference. What type of document is this? Check the MDPI guide for authors to see if the year should be in bold for this type of reference.

18. Please include the doi.

20. Please include the page number and the doi.

29. Please include the doi.

30.  Check the MDPI guide for authors to see if the year should be in bold for this type of reference.

42. Check the MDPI guide for authors to see if the year should be in bold for this type of reference.

Author Response

Academic editor’s comment

The system of healthcare and GP is different among countries. Kindly add brief explanation of French GP system to the authors' response after the first-round review.

Thank you for reminding us to clarify the French healthcare context so that our data is understood by all your international readership. In the submitted version of the manuscript, we wrote a lengthy section about the French healthcare system, for the very reason you mentioned. It can be found between lines 57 and 71 (pasted below). We thought that this would be sufficient to understand the context and applicability (or adaptations needed) in other healthcare contexts. We are available to discuss this point furthermore if you still judge this necessary.

“The French primary healthcare sector is mainly composed of private-sector healthcare professionals that are paid according to a fee-for-service model, financed by the national health insurance system. Since 2006, a gate-keeping system to regulate access to specialist care has been implemented in France [8]. During the same period, we also observed a trend towards group practices in primary care [9]. More recently, two main types of multidisciplinary primary-care teams coordinated around a health project have emerged: independent multidisciplinary groups (1,617 centers), composed of several independent healthcare professionals, and healthcare centers (428 centers), where healthcare professionals are employed. Currently, in France, about 15% of primary care professionals (depending on the profession) practice in these primary-care teams. In 2016, a new territorial organization framework (“territorial and professional healthcare communities”) was introduced to encourage primary care actors to assume collective social responsibility in their region [10-12] This is quite similar to the primary care clusters in Wales [13]. Today, the French primary care sector is evolving towards more collective and integrative practices.”

Reviewer 1

This is a clear account of a well-constructed survey of metropolitan French GPs during the first lockdown of COVID-19 between 17 March 2020 and 11 May 2020 to determine the result of the lockdown on their practice and on the ways in which they attended to patients.

There are, however, important details that have been left out, including some missing references, and a few small errors to correct. These details, references and the errors are noted in the line-by-line suggested edits to follow. Once these corrections are made, this study represents one that could be easily replicated because of the care taken in devising and implementing the study as well as conducting the analysis.

We thank the reviewer for the time they took to perform a thorough review of our manuscript and the comments provided. Please find below our point-by-point response.

Line by line suggested edits:

32 Change “~5%” to “5.8%”. The rest of the percentages quoted are exact. This one should be too.

Done

44 Although, at this time, the world understands what a COVID lockdown represents, for future generations, please add information indicating what is COVID and why there would be a lockdown. Please reference this as well.

Modified as follows, “In response to the Severe acute respiratory syndrome-coronavirus 2 (SARS-CoV-2)-related pandemic, the French government implemented the first lockdown between the 17th of March 2020 and the 11th of May 2020. Leaving home was only allowed for a limited list of compelling reasons, leading to a major limitation of all activities in France. As in other countries worldwide, the aim of the lockdown was to containing the epidemic during this time when there was much uncertainty and a scarcity of critical material [1]. Initially during the lockdown, the management of COVID-19 (coronavirus disease 2019) was mainly supported (…)” and we added 2 references: 1 for the pandemic, the other for the lockdown.

89 Please indicate the disciplines represented in this interdisciplinary research group.

Thank you, we have updated the manuscript (line 95-97) to indicate that health professionals from that included experts from general practice, public health, health services research, advanced nursing, midwifery, sociology and representants of multidisciplinary practices.

91-92 How were these seven primary healthcare experts chosen and contacted. What was their reason for participating in the review?

These experts were members of the research group and five out of them are the authors of this paper (the two others were not involved in the subsequent phases of the study).

92 Please provide some details about the pilot study.

We added “This pilot was aimed at evaluating the understanding and readability of the questions and of the answers’ modalities. The research group reviewed the questionnaire according to the pilot’s feedback until consensus about improvement was reached.”

101 How was the survey constructed? What previous surveys influenced the design of this survey?

The questionnaire was constructed using mostly closed questions, as is standard practice when constructing quantitative questionnaires. For the multiple-choice questions, the experience of the professionals involved in the construction of the questionnaire guided the decisions about the most relevant modalities to retain. As previously mentioned in the article, some questions or modalities were inspired by the first survey our research team conducted in March 2020. No other questionnaire dedicated to primary care organization during the COVID crisis was available at the time of designing our questionnaire.

107 Change “patients that usually” to “patients that they usually”.

Done

110 When additional detail is added about COVID in line 44, be sure to include that COVID-19 is SARS-CoV-2.

Done, see previous comment.

120 Delete “GP, general practitioner.” This information is unnecessary.

Done

124 Please provide information regarding why there might be duplicates.

We added: “duplicates appeared due to more than one completion of the questionnaire by some participant”. It may have appeared for examples if they received the survey link several times through different ways (personal e-mail, professional e-mail, social network, etc) or if they were interrupted during the initial filling of the survey then connected again to the survey.

127-128 Please state why only data from GPs working in metropolitan France were included.

Due to “significant differences between metropolitan France and overseas territories in terms of 1) health situation and 2) health care system organization”, it was not relevant to analyse all answers altogether. Only 28 survey respondents were located in overseas territories (2 in Guadeloupe, 6 in Martinique, 18 in Réunion and 2 in Mayotte), these responses were not sufficient to perform a specific analysis. Finally, due to the sampling strategy, it was not possible to limit the administration of the questionnaire to metropolitan GPs only; so, we decided not to include these 28 observations.

133 Please provide a recent reference to scree plots and Kaiser criterion. 

We have added the following reference:

Braeken J, van Assen MALM. An empirical Kaiser criterion. Psychol Methods. 2017 Sep;22(3):450-466. doi: 10.1037/met0000074. Epub 2016 Mar 31. PMID: 27031883.

134 Change “explains” to “explain”.

Done

136 Please provide a recent reference to the Ward minimum-variance linkage method.

We added the following reference that presents different methods to construct dendograms.

Everitt, B. S., Landau, S. and Leese, M. (2001), Cluster Analysis, 4th Edition, Oxford University Press, Inc., New York; Arnold, London.

150 Please provide a recent refence for the Wald tests.

We added the following reference:

Fahrmeir, Ludwig; Kneib, Thomas; Lang, Stefan; Marx, Brian (2013). Regression: Models, Methods and Applications. Berlin: Springer. p. 663.

161 To correspond with the data in Table 2 (and to add up to 100%), please change “44.2%” to “44.8%”. Regarding working less frequently alone, looking at Table 2, 15.5% represents those GP who worked alone, not those who didn’t work alone. As well, the 15.5% versus 39% is between those who answered the survey and all GPs in France rather than those who didn’t work alone and those who did. If the authors want to make this comparison with all French GPs, rather than in relation to whether those surveyed in the study work alone, then they must state this is what they are comparing. If the point they want to make is that most metropolitan French GPs who don’t work alone were those that they surveyed they must add 43.4% to 41.2% to indicate that 84.6% did not work alone.

The comparisons were between the responders to the survey and all the GPs in France. This is why 55.2% (survey responders) is compared to 44.2% (all GPs in France) and 15.5% (survey responders) is compared to 39.0% (all GPs in France). We added a few words to make the sentence clearer.

163 If all French regions were represented how does this relate to the point made in lines 127-128 that only those GPs working in metropolitan France were included?

Thanks for this remark, we added “metropolitan” (i.e., “All metropolitan French regions were represented in the study”).

Table 3. Change “[40-55[“ to “[40-55]” and Change “≥55” to “>55”.

Thanks for your remark. In fact, the categories were: less than 40 years (<40); 40 to 54 years ([40-54]); 55 years or more (≥55). Therefore, we have changed “[40-55[“ to “[40-54]” throughout the manuscript. We kept “≥55” because that one was correct. We also changed the category “[20-30[“ patients/day by “20 to 29” patients/day.

It is interesting that the largest percentage of GPs >55 were the ones who used the hospital cluster (Cluster 3). Is it a greater benefit to be able to use the hospital cluster and is this why they may have selected this type of practice? Please comment on this in the manuscript.

The survey question related to “referrals of outpatients with suspected COVID-19” was limited to patients who didn’t need specialized care either for diagnoses or therapeutic purposes. We don’t think that, for these patients, that it is necessary to use hospital services, because of the higher costs of hospital care and because of the limited resources of overburdened hospitals at the time of the lockdown.

When looking at Table 3 (univariate), percentage of GPs ≥ 55 years was indeed “as high as” 57.8% in the cluster 3 while it was “only” between 30% and 33% in the 3 others clusters. However, in the multivariate analyses, the association between age and cluster 3 was just borderline statistically significant (aOR = 1.50, CI95 =0.98-2.28, p=0.05). One reason for that is that in France older GPs work more often alone (solo practice). There was some confusion bias, and we believe that the most important factor was the type/size of practice. The association between the type/size of practice and cluster 3 (aOR = 1.98, CI95 =1.27-3.11, p≤0.01) was high after adjusting for other variables, including age, that is why we chose to comment on type/size of practice rather than on age in our discussion, as following: “Our study suggests that the organization features (practice type and size) were the factors more significantly associated with the GPs’ capacity for adaptation. GPs working alone relied more on hospitals, suggesting that they had fewer resources to reorganize their work within the primary care framework. The lower capacity of GPs working in small practices to adapt was also identified in the United States [39] while task changes were larger in primary care practices employing a wider range of professionals, based on a large survey undertaken in 2020-2021 in 38 countries [40].”

Why do the authors make a point of differentiating COVID-19 from SARS-CoV-2 in their list of variables?

We hope that we correctly understood your question… In our survey we considered separately “Being at risk of severe COVID-19” and “Fear of SARS-CoV-2” as following:

“Being at risk of severe COVID-19” related to the GP’s self-assessment of his/her own personal risk of developing a severe COVID-19 in the case he/she gets infected by the SARS-CoV-2 (at that time when there was no vaccine available in France). This assessment was expected to be mainly based on the well-known list of risk factors for severe COVID-19: age, obesity, chronic diseases…. The exact survey question was: Are you personally at risk for a severe form of COVID-19?

“Fear of SARS-CoV-2” related to the GP’s fear of catching the virus. Our intention was to explore more subjective aspects of the GP’s personal experience during that special time of pandemics. Our hypothesize was that a GP may have been objectively at risk of severe COVID but not be afraid, while the contrary could also exist (not being objectively at risk but being afraid). The exact survey question was: Are you currently afraid to catch the virus?

Was “>2” the only option on the survey for the number of patients who died if the answer was > 2? Is it possible that some GPs had a significantly more deaths than two? If so, how was this captured in the survey? If it was not, why was this not considered important? Please explain in the manuscript.

In the questionnaire, the question didn’t mention any categories, it was an open question requiring a quantitative discrete answer.

The preliminary analyses showed the following distribution: 0 dead patient = 71.6% of the sample / 1 dead patient = 15.4%. / 2 dead patients = 5.8% / 3 dead patients = 3.1% / all the rest of the valid responses (>3 dead patients) = 4.0% of the sample (12 modalities of response, representing each 0.1 to 1.2% of the sample). Due to the asymmetric distribution of the variable, we chose to recode it into a three-modality variable which was a compromise between clinical significance and each modality sub-sample size. Some GPs had significantly more than two regular patients who died from COVID-19 but this situation was quite rare and it was not possible to consider it as a modality for quantitative analyses. We added a sentence at the end of the sample characteristics paragraph: “Regarding the number of each GP’s regular patients who died due to COVID-19, the very asymmetric distribution of the data led to recode it into 3 modalities: modalities: 0 patient (70.971.6% of the sample); 1 or 2 (21.03%); 3 or more patients (7.1%).”

We thank you very much for your question because it led us to see that there were some mistakes – the previous figures were that of patients hospitalized for COVID-19, not died from - about this variable in the Table 3 (univariate analysis) that we have corrected this in the revised manuscript.

Again, it is noted that those GPs in Cluster 3 had the highest percentage related to those who practiced alone in comparison with the other clusters. Is this because they were generally older than the GPs in the other clusters and older GPs were more likely to work alone?

Yes, see previous comment. Our analyses suggest that there was some confusion bias in univariate analyses and that working alone may be more strongly associated with cluster 3 than being older than 55 years.

191 Delete “GPs, general practitioners; SARS-COV-2, severe acute respiratory syndrome coronavirus 2.” This information is unnecessary.

Done

192 Given that there was a range that GPs could select from, what does the percentage indicated represent? Is it GPs who selected anything other than “Never”? Please explain in the manuscript. 195 Given that there was a range that GPs could select from, what does the percentage indicated represent? Is it the GPs who selected anything other than “Not at all”? Please explain in the manuscript.

Thanks for this request for clarification. We added below Tables 3 and 4: “Often” or “Always” clustered as “Yes” versus “Never” or “Rarely” clustered as “No”; and “Are you afraid of being infected?” Answers “Not at all” or “slightly” clustered as “No” versus “Afraid” or “very afraid” clustered as “Yes”. We hope it is now clearer.

213 Also note that these GPs were more frequently >55.

See previous comments on association between age / working alone and cluster 3

Table 4. Change “[40-55[“ to “[40-55]” and Change “≥55” to “>55”.

As previously answered, we changed to “[40-54]” and we did not change “≥55” because it was correct.

227 Change “. Level” to “Level”

Done

266-272 The authors have commented on younger GPs but have not made a general comment on those mid-careers or those who are older GPs. Mention should be made of results associated with them as well. 

We added some sentences in the discussion “Contrastingly, older GPs more often used hospitals resources. When adjusting for other variables, this association remained just borderline significant, suggesting some confusion bias. Specifically, this observation may be (at least partly) explained by the fact that older GPs more often work alone.”

275-276 It may also suggest that they needed few resources in working alone because they had more hospital privileges than other GPs—there was no need for duplication. Please comment on this in the manuscript.

We are not sure to understand what the reviewer means here. What does “hospital privileges” mean? Does it apply to the French context? As previously mentioned, we don’t think that it was of greater value to use hospital services for patients having non severe COVID-19, because of unnecessarily higher costs of hospital care and because of the limited resources of overburdened hospitals at that time.

Supplementary material Table S2

Rather than putting “GPs, general practitioners” at the bottom of the table, in the title, change “general practitioners’” to “general practitioners’ (GPs)”.

Done

References

Generally, remember that MDPI journals require short forms for journal names rather than the full name of the journal. Some of the references need to have the journal title reduced to its short form.

  1. This is an incomplete reference. What type of document is this?

Updated

  1. This is an incomplete reference. What type of document is this? Check the MDPI guide for authors to see if the year should be in bold for this type of reference.

Updated

  1. Please include the doi.

A doi is not available for this document.

  1. Please include the page number and the doi.

Updated

  1. Please include the doi.

Updated

  1. Check the MDPI guide for authors to see if the year should be in bold for this type of reference.

Updated

  1. Check the MDPI guide for authors to see if the year should be in bold for this type of reference.

Updated

Reviewer 2 Report

Firstly, I would like to congratulate the authors for creating this great and important study!

Subdomains were very thoughtfully designed, emphasizing the key elements among health care system levels, and it's functioning. The authors highlighted the important limits of this study (such as overrepresented teaching GP's, and the possibility of the social desirability bias) which are crucial for the interpretation of the results. One of the most intriguing conclusions from the discussion area was that the COVID-19 pandemic did not result in organizational rupture but in a strengthening of existing structures, partnerships, and dynamics, which is very important in light of all chronic diseases and care for those patients during the pandemic. In many European countries, especially in Eastern Europe, primary health care level is still disorganized and underestimated, leading to unnecessary and unjustified costs due to the use of general hospitals and tertiary centers. We need to do a similar survey in other European countries to compare different health systems and improve its functionality. Studies like this done on primary practices are very rare in general and could significantly change the perception of leading structures towards primary health care level. Accordingly, this study highlights a very important conclusion, which I hope will be read and cited worldwide.

Because of the aforementioned, this study should be published in its current form.

Author Response

Reviewer 2

Firstly, I would like to congratulate the authors for creating this great and important study!

Subdomains were very thoughtfully designed, emphasizing the key elements among health care system levels, and it's functioning. The authors highlighted the important limits of this study (such as overrepresented teaching GP's, and the possibility of the social desirability bias) which are crucial for the interpretation of the results. One of the most intriguing conclusions from the discussion area was that the COVID-19 pandemic did not result in organizational rupture but in a strengthening of existing structures, partnerships, and dynamics, which is very important in light of all chronic diseases and care for those patients during the pandemic. In many European countries, especially in Eastern Europe, primary health care level is still disorganized and underestimated, leading to unnecessary and unjustified costs due to the use of general hospitals and tertiary centers. We need to do a similar survey in other European countries to compare different health systems and improve its functionality. Studies like this done on primary practices are very rare in general and could significantly change the perception of leading structures towards primary health care level. Accordingly, this study highlights a very important conclusion, which I hope will be read and cited worldwide.

Because of the aforementioned, this study should be published in its current form.

We would like to thankthe reviewer for the time taken to evaluate our work, and emphasizing one of our key messages. We do share your conclusions and hope to make a difference, contributing paper after paper to better primary healthcare systems offered to populations worldwide.

Reviewer 3 Report

The study shows the potential adaptation of family medicine practices to new challenges in primary care. Analyzing the adaptations and not only making suggestions for practices based on theoretical considerations is, in my view, a new innovative approach that should be researched further. 

I have no suggestions for changes to the manuscript.

Author Response

Reviewer 3

The study shows the potential adaptation of family medicine practices to new challenges in primary care. Analyzing the adaptations and not only making suggestions for practices based on theoretical considerations is, in my view, a new innovative approach that should be researched further. 

I have no suggestions for changes to the manuscript.

We thank the reviewer for the time taken to review our work, and for the favourable comment about our analytical approach.
